

# Reverse engineering approach for improving the quality of mobile applications

Eman K. Elsayed[1], Kamal A. ElDahshan[2], Enas E. El-Sharawy[1,3] and Naglaa E. Ghannam[1]

[1] Department of Mathematical and Computer Science, Faculty of Science, Al-Azhar University, (Girls Branch), Cairo, Egypt
[2] Department of Mathematical and Computer Science, Faculty of Science, Al-Azhar University, Cairo, Egypt
[3] Computer Department, College of Science and Humanities in Jubail, Imam Abdulrahman Bin Faisal University, Kingdom of Saudi Arabia

Corresponding author
Naglaa E. Ghannam,
naglaasaeed@azhar.edu.eg

## ABSTRACT

**Background:** Portable-devices applications (Android applications) are becoming complex software systems that must be developed quickly and continuously evolved to fit new user requirements and execution contexts. Applications must be produced rapidly and advance persistently in order to fit new client requirements and execution settings. However, catering to these imperatives may bring about poor outline decisions on design choices, known as anti-patterns, which may possibly corrupt programming quality and execution. Thus, the automatic detection of anti-patterns is a vital process that facilitates both maintenance and evolution tasks. Additionally, it guides developers to refactor their applications and consequently enhance their quality.

**Methods:** We proposed a general method to detect mobile applications' anti-patterns that can detect both semantic and structural design anti-patterns. The proposed method is via reverse-engineering and ontology by using a UML modeling environment, an OWL ontology-based platform and ontology-driven conceptual modeling. We present and test a new method that generates the OWL ontology of mobile applications and analyzes the relationships among object-oriented anti-patterns and offer methods to resolve the anti-patterns by detecting and treating 15 different design's semantic and structural anti-patterns that occurred in analyzing of 29 mobile applications. We choose 29 mobile applications randomly. Selecting a browser is not a criterion in this method because the proposed method is applied on a design level. We demonstrate a semantic integration method to reduce the incidence of anti-patterns using the ontology merging on mobile applications.

**Results:** The proposed method detected 15 semantic and structural design anti-patterns which have appeared 1,262 times in a random sample of 29 mobile applications. The proposed method introduced a new classification of the anti-patterns divided into four groups. "The anti-patterns in the class group" is the most group that has the maximum occurrences of anti-patterns and "The anti-patterns in the operation group" is the smallest one that has the minimum occurrences of the anti-patterns which are detected by the proposed method. The results also showed the correlation between the selected tools which we used as Modelio, the Protégé platform, and the OLED editor of the OntoUML. The results showed that there was a high positive relation between Modelio and Protégé which implies that the combination between both increases the accuracy level of the detection of anti-patterns. In the evaluation and

analyzing the suitable integration method, we applied the different methods on homogeneous mobile applications and found that using ontology increased the detection percentage approximately by 11.3% in addition to guaranteed consistency.

# INTRODUCTION

Mobile applications take center stage in our lives today. We utilize them anywhere, at any time and for everything. We use them to peruse websites, shop, search for everything we need and for basic administration such as banking. For the importance of mobile applications, their reliability and quality are critical. Like any other applications, the initial design of mobile applications is affected by bug-settling and the introduction of new properties, which change the initial design; this can occasionally affect the quality of design (*Parnas, 1994*). This aspect is known as software degeneration, which can exist in the form of design flaws or anti-patterns (*Eick et al., 2001*).

One of the most important factors in the development of software systems is improving software quality. The success of software design depends on the availability of quality elements such as maintainability, manageability, testability, and performance. These elements are adversely affected by anti-patterns (*Afjehei, Chen & Tsantalis, 2019*; *Yamashita & Moonen, 2013*). Anti-patterns are bad practice in software design. The automatic detection of anti-patterns is a good way to support maintenance, uncomplicate evolution tasks, and improve usability. In addition to the general advantages of detecting anti-patterns, we think that detecting anti-patterns provides developers with a way to ensure that the detected anti-patterns will not be repeated in applications revisions. Also, detecting anti-patterns may improve both operational characteristics and user experience.

We note that there are many other approaches interested in detecting anti-patterns in the code level as introduced by *Morales et al. (2016)* and *Alharbi et al. (2014)*. However, it has been noted that anti-pattern detection at the design level reduces many code anti-patterns and is more general.

According to *Raja (2008)*, engineering is the process of designing, manufacturing, assembling, and maintaining products and systems. Engineering has two types, forward engineering, and reverse engineering (RE) as presented by *Raja (2008)*. *Chikofsky & Cross (1990)* defined RE as the process of analyzing software systems to identify the components of the systems and the interrelationships between them and presenting the systems in other forms or at a higher level of abstraction. The term RE according to our approach, refers to the process of generating UML diagrams followed by generating OWL ontologies of mobile applications through importing and analyzing the bytecode.

Generally, we can use ontology re-engineering for direct incorporation as an Ontology development method (*Obrst et al., 2014*) by allowing the designer to analyze the common components dependence.

Designing a pattern of mobile application remains an ongoing research challenge. The proposed approach aims to detect structural and semantic anti-patterns in the design of mobile applications as well as to show which method is better for the integration of applications.

Motivated by the research mentioned above, the major contributions of this paper are sixfold:

- Presenting a new method for generating OWL ontology of mobile applications.
- Presenting a general method for enhancing the design of a pattern of a mobile application.
- Illustrating how the proposed method can detect both structural and semantic anti-patterns in the design of mobile applications.
- Describing how we evaluate the proposed method in 29 mobile applications. Showing how it detects and treats 15 designs' semantic and structural anti-patterns that appeared 1,262 times.
- Showing how semantic integration among mobile applications decreases the occurrences of anti-patterns in the generated mobile application pattern.
- Analyzing the relationships among the object-oriented anti-patterns and the detection tools.

In the rest of the paper, we subsequently present the related work. Next, we present some basic definitions, and the details of the proposed approach is described. After that, the empirical validations of the proposed method are presented, followed by the results and discussion. Finally, the concluding remarks are given, along with scope for future work.

## RELATED WORKS

Many empirical studies have demonstrated the negative impact of anti-patterns on change-proneness, fault-proneness, and energy efficiency (*Romano et al., 2012*; *Khomh et al., 2012*; *Morales et al., 2016*). In addition to that, *Hecht et al. (2015a)*, *Chatzigeorgiou & Manakos (2010)*, *Hecht, Moha & Rouvoy (2016)* observed an improvement in the user interface and memory performance of mobile apps when correcting Android anti-patterns. They found that anti-patterns were prevalent in the evolution of mobile applications. They also confirmed that anti-patterns tend to remain in systems through several releases unless a major change is performed on the system. Many efficient approaches have been proposed in the literature to detect mobile applications' anti-patterns.

Some researchers concentrate on ensuring that the soft is free of contradictions which are called consistency. *Alharbi et al. (2014)* detected the anti-patterns related to inconsistency in mobile applications that were only related to camera permissions and similarities. *Joorabchi, Ali & Mesbah (2015)* detected the anti-patterns related to inconsistency in mobile applications using a tool called CHECKCAMP that was able to detect 32 anti-patterns related to inconsistencies between application versions. *Hecht et al. (2015b)* used the Paprika approach to detect some popular object-oriented anti-patterns in

the code of mobile applications using threshold technique. *Linares-Vásquez et al. (2014)* detected 18 object oriented (OO) anti-patterns in 1,343 Java mobile applications by using DÉCOR. This study focused on the relationship between smell anti-patterns and application domain. Also, they showed that the presence of anti-patterns negatively impacts software quality metrics; in particular, metrics related to fault-proneness. *Yus & Pappachan (2015)* analyzed more than 400 semantic Web papers, and they found that more than 36 mobile applications are semantic mobile applications. They showed that the existence of semantic helps in better local storage and battery consumption. The detection of semantic anti-patterns will improve the quality of mobile applications. *Palomba et al. (2017)* proposed an automated tool called A DOCTOR. This tool can identify 15 Android code smells. They made an empirical study conducted on the source code of 18 Android applications and revealed that the proposed tool reached 98% precision and 98% recall. A DOCTOR detected almost all the code smell instances existing in Android applications. *Hecht et al. (2015b)* introduced the PAPRIKA tool to monitor the evolution of mobile application quality based on anti-patterns. They detected the common anti-patterns in the code of the analyzed applications. They detected seven anti-patterns; three of them were OO anti-patterns and four are mobile applications anti-patterns.

Reverse engineering is the process of analyzing software systems to identify the components of the systems and the interrelationships between them and presenting the systems in other forms or at a higher level of abstraction (*Chikofsky & Cross, 1990*).

In this paper, we used RE to transfer code level to design level for detecting mobile applications' anti-patterns. RE techniques are important for understanding the construction of the user interface and algorithms of applications. Additionally, we can know all the properties of the application, its activities, and permissions and can read the Mainfest.xml of the applications. RE techniques have been used with mobile applications for many purposes not just for detecting anti-patterns. *Song et al. (2017)* used RE for improving the security of Android applications. While *Zhou et al. (2018)* used the RE technique to detect logging classes and to remove logging calls and unnecessary instructions. Also, *Arnatovich et al. (2018)* used RE to perform program analysis on a textual form of the executable source and to represent it with an intermediate language (IL). This IL has been introduced to represent applications executable Dalvik (dex) bytecode in a human-readable form.

## ONTOLOGY AND SOFTWARE ENGINEERING

According to the *IEEE Standard Glossary of Software Engineering Terminology-Description (1990)*, software engineering is defined as "the application of a systematic, disciplined, quantifiable approach to the development, operation, and maintenance of software."

Also, from the knowledge engineering community perspective, computational ontology is defined as "explicit specifications of a conceptualization." According to *Calero, Ruiz & Piattini (2006)*, *Happel & Seedorf (2006)*, the importance of sharing knowledge to move the software to more advanced levels require an explicit definition to help machines interpret this knowledge. *Happel & Seedorf (2006)* decided that ontology is the most promising way to address software engineering problems.

*Elsayed et al. (2016)* proofed the similarities in infrastructures between UML and ontology components. They proposed checking some UML quality features using ontology and ontology reasoning services to check consistency and redundancies over UML models. This would lead to a strong relationship between software design and Ontology development.

In software engineering, ontologies have a wide range of applications, including model transformations, cloud security engineering, decision support, search, and semantic integration (*Kappel et al., 2006*; *Aljawarneh, Alawneh & Jaradat, 2017*; *Maurice et al., 2017*; *Bartussek et al., 2018*; *De Giacomo et al., 2018*). Semantic integration is the process of merging the semantic contents of multiple ontologies. The integration may be between applications that have the same domain or have different domains to take the properties of both applications. We make ontology integration for many reasons: to reuse the existing semantic content of applications, to reduce effort and cost, to improve the quality of the source content or the content itself, and to fulfill user requirements that the original ontology does not satisfy.

## PROPOSED METHOD

In this section, we introduce the key components of the proposed method for analyzing the design of mobile applications to detect design anti-patterns, and for making semantic integration between mobile applications via ontology reengineering.

The proposed method for anti-pattern detection consists of three main phases and is summarized in Fig. 1. Also, there is an optional phase called the integration phase.

1. **The first phase** presents the process of reformatting the mobile application to Java format.
2. **The second phase** presents the reverse-engineering process. In this phase, we used RE to reverse the Java code of mobile applications and generating UML class diagram models. Additionally, many design anti-patterns were detected. The presented reverse approach is accurate enough to analysis the information that we need about APK to reverse UML models of the applications.
3. **The third phase** completes the anti-patterns detection and correction processes. This phase converts UML mobile application model to OWL ontology, then analyzes the relationships among object-oriented anti-patterns and offers methods to resolve the anti-patterns related to semantic and inconsistency. After that, we can regenerate the Java code of mobile applications. The developer can ensure that anti-patterns in existing applications will not be repeated in application revisions and may improve both operational characteristics and user experience.
4. **The integration phase** is an optional fourth phase. In this phase, we integrate two applications by merging the OWL ontologies of both applications. From these two ontologies, we will yield one integrated application for doing both services with minimum anti-patterns.

We will present in detail the rationale provided for why this integration is needed as an optional phase if we need.

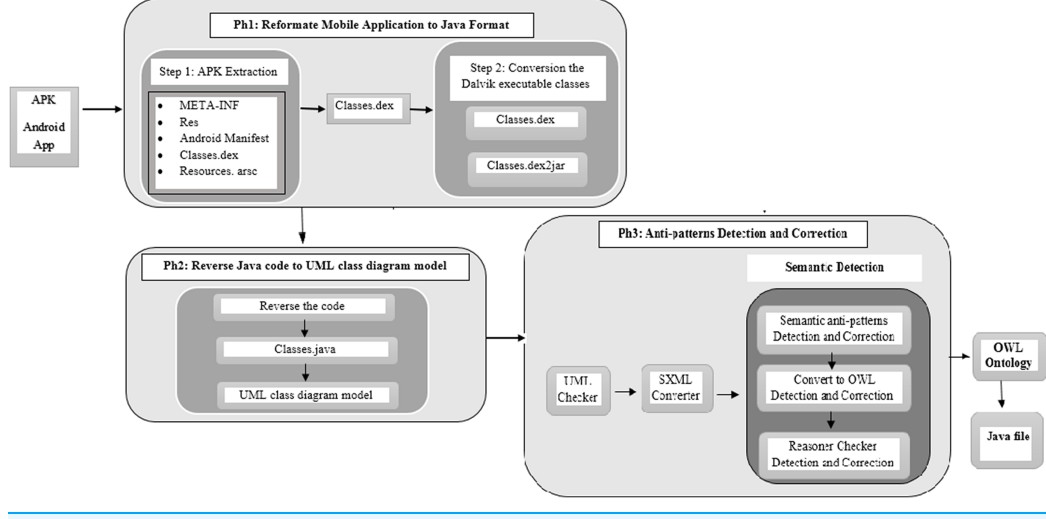

**Figure 1 The proposed method phases.**

## The integration of mobile applications

The integration process is most for the inclusion of new skill sets for applications such as IOT or monitoring applications or potentially voice-activation integration into an existing application. But, here we were interested in presenting a new manner for homogenous integration to combine the advantages of two mobile applications in a new pattern. In this section, we provided a rationale for why this integration is needed and presenting the integration as an extra phase if we need where the other detection phases do not change. Patterns are advanced methods to develop any mobile applications. The integration or merging of mobile applications is a good step in mobile application development. The advantage of the integration of mobile applications is in responding to the puzzling selection of the appropriate application from a set of applications. This will achieve the same objective if each application has a different advantage and the developer wants to start to improve pattern combines all advantage without anti-patterns.

To clear our idea, we choose two homogenous applications: Viber and WhatsApp. They are the most popular messaging and Voice Over IP applications. Both Viber and WhatsApp are similar in services, features, security, and cost. There is plenty to like about both applications: they produce the same services as end-to-end encryption, support groups and video calls, support on any operating system, allow transmission of documents or multimedia, and work over 3G, 4G, and Wi-Fi. Well, both are fantastic in their way, but which one is better for the developer as a pattern for refinement? We found that Viber had been offering both video and voice calling for a far longer time than WhatsApp and has a hidden chat feature. Also, Viber permits the user to play a list of games with other Viber contacts. However, WhatsApp is popular and easy to use. We can make the integration of both applications and take the best skills of both.

We imagine that when producing a new application we can directly integrate it to the old one without replacing.

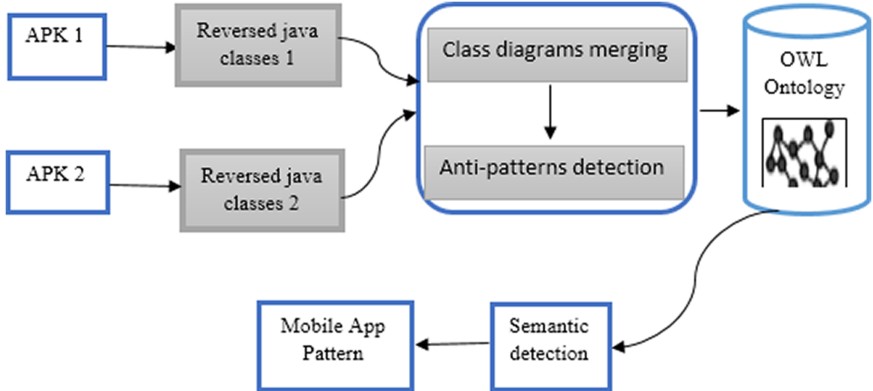

**Figure 2** **Merging UML class diagrams of the mobile apps.**

In the case of heterogonous integration applications, the developer, for example, may want to develop a new health care hybrid application. From the website "free apps for me" (https://freeappsforme.com/), a developer can find at least seven applications for measuring blood pressure. All of them are free and available on a different platform. There are also at least 13 diabetes applications. When a developer merge two applications (one for measuring blood pressure as the "Smart Blood Pressure" application and the other for controlling diabetes as the "OneTouch Reveal" application), the integration phase will yield one integrated application for doing both services, with minimum anti-patterns. Then the developer can add the new relations between these disease controller without conflict.

The integration allows the combination of the skills of both applications to get new mobile application pattern. These two examples of two types of integration answer the question of why we need to integrate mobile applications.

We suggest using the integration pattern, then comparing between the two integration proposed methods to select the suitable one.

The first integration method is for after decompiling the APK of the applications. We use RE methodology for generating one UML class diagram of both applications. Then we start the detection of the anti-patterns process for the integrated application (Fig. 2).

The second integration method is through merging the OWL ontologies of both applications using the Prompt plugin in protégé as the ontology editor as introduced in Fig. 3.

## The implementation

In this section, we propose the implementation of the proposed detection method and determine which packages are suitable for each phase.

- **The first phase:** APK files are zip files used for the installation of mobile apps. We used the unzip utility for extracting the files stored inside the APK. It contained the AndroidManifest.xml, classes.dex containing the Java classes we used in the reverse process, and resources.arsc containing the meta-information. We de-compiled the APK files using apktool or Android de-compiler. Android de-compiler is a script that combines different tools to successfully de-compile any (APK) to its Java source code

and resources. Finally, we used a Java de-compiler tool such as JD-GUI to de-compile the Java classes. JD-GUI is a standalone graphical utility that displays the Java code of ". class" files. The input of the first phase was the APK file of the mobile application and the output was the Java classes of the APK application. JD-GUI is accurately enough to generate the Java code that we use to reverse the models of the applications.

- **The second phase:** We used a RE approach for generating the UML class diagram models of the mobile applications. *Elsayed, El-Dahshan & Ghannam (2019)* compared between UML tools, the authors found that Modelio 3.6 is a suitable tool for modeling and detecting UML design anti-patterns. The UML class diagram was generated by reversing the Java binaries of the mobile app. Detecting anti-patterns in the UML model is the first step in the detection process. The input of the second phase was classes.java of the app and the output was the UML class diagram model of the app with a list of the detected anti-patterns.

- **The third phase:** By converting the model to XML format, we could generate it as an OntoUML model in OLED, which is the editor of OntoUML for detecting semantic anti-patterns. OntoUML is a pattern-based and ontologically well-founded version of UML. Its meta-model has been designed in compliance with the ontological distinctions of a well-grounded theory named the unified foundational ontology. OLED editor also supports the transformation of the OLED file to the OWL ontology of the mobile app, allowing the detection of inconsistency and semantic anti-patterns using the "reasoner" ontology in Protégé. Protégé is the broad ontology editor commonly used by many users.

**The integration phase (the fourth optional phase):** we propose two methods for integrating mobile applications. The first method is merging the UML models at the second phase when we reverse the models from Java code and then completing the detection phases over the integrated application. The second method is merging the OWL ontologies of the both applications using a Prompt (Protégé plugin) to generate one OWL ontology pattern. Figure 4 shows the both applications "Viber and WhatsApp" components before merging. Figure 5 shows the integrated application; Fig. 5 has three tabs (classes, slots, and instances) which are the components of the ontology. Every tab shows the components of its type after integration. Finally, we used "Reasoner in Protégé" to check the consistency after integration.

## EMPIRICAL VALIDATIONS

We assessed our approach by reporting the results we obtained for the detection of 15 anti-patterns on a random sample of 29 popular Android applications downloaded randomly from the APK Mirror.

### Applications under analysis

Table 1 presents the downloaded applications from the APK Mirror. We selected some popular applications such as YouTube, WhatsApp, Play Store, and Twitter. The size of the applications included the resources of the application, as well as images and data files

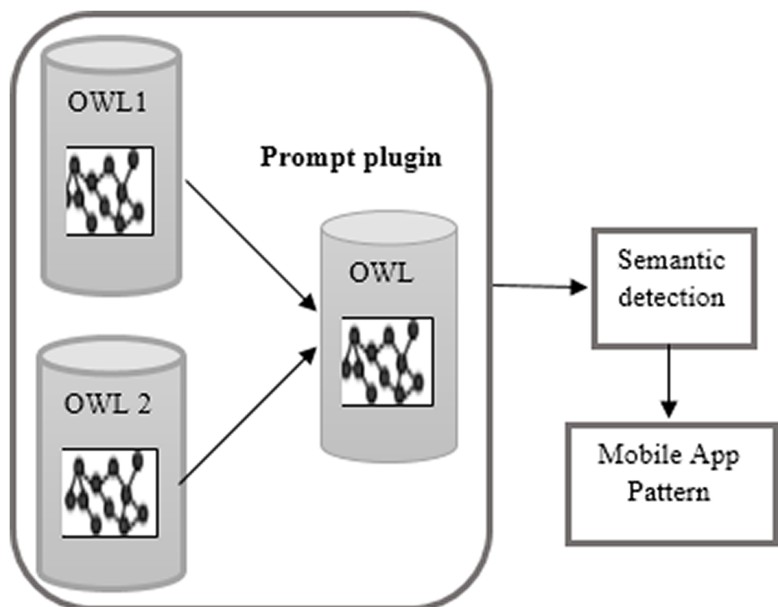

**Figure 3** OWL ontology merging.

| Name | Arg1 | Arg2 |
|------|------|------|
| merge | ● http://nemo.inf.ufes.br/viber3.owl#IndividualCo | ● whatsappfinal:IndividualConcept *whatsapp* |
| merge | ● viber3:TimeSlice *viber* | ● whatsappfinal:TimeSlice *whatsapp* |
| merge | ● viber3:TemporalExtent *viber* | ● whatsappfinal:TemporalExtent *whatsapp* |
| merge | ● viber3:Object *viber* | ● whatsappfinal:Object *whatsapp* |
| merge | ● viber3:Moment *viber* | ● whatsappfinal:Moment *whatsapp* |
| merge | ● viber3:Moment *viber* | ● whatsappfinal:Mode *whatsapp* |
| merge | ● viber3:FunctionalComplex *viber* | ● whatsappfinal:FunctionalComplex *whatsapp* |
| merge | ● viber3:Collective *viber* | ● whatsappfinal:Collective *whatsapp* |
| merge | ● viber3:Collective *viber* | ● whatsappfinal:CollectionsTS *whatsapp* |
| merge | ● viber3:Quantity *viber* | ● whatsappfinal:Quantity *whatsapp* |
| merge | ● viber3:FunctionalComplexTS *viber* | ● whatsappfinal:FunctionalComplexTS *whatsap|* |
| merge | ● viber3:CollectiveTS *viber* | ● whatsappfinal:CollectiveTS *whatsapp* |
| merge | ● viber3:CollectiveTS *viber* | ● whatsappfinal:CollectionsTS *whatsapp* |
| merge | ● viber3:QuantityTS *viber* | ● whatsappfinal:QuantityTS *whatsapp* |
| merge | ● viber3:Relator *viber* | ● whatsappfinal:Relator *whatsapp* |
| merge | ● viber3:RelatorTS *viber* | ● whatsappfinal:RelatorTS *whatsapp* |
| merge | ● viber3:Mode *viber* | ● whatsappfinal:Moment *whatsapp* |
| merge | ● viber3:Mode *viber* | ● whatsappfinal:Mode *whatsapp* |
| merge | ● viber3:ModeTS *viber* | ● whatsappfinal:ModeTS *whatsapp* |

**Figure 4** "Viber and Whatsapp" ontologies before integration in Protégé.

(Table 1). The research study included the identification and repetition of anti-patterns across different domains and different sizes.

## Case study on "Avast Android Mobile Security"

To explain the proposed method, we presented a snapshot of it in a different case study "Avast Android Mobile Security." The case study is one of the 29 mobile applications that

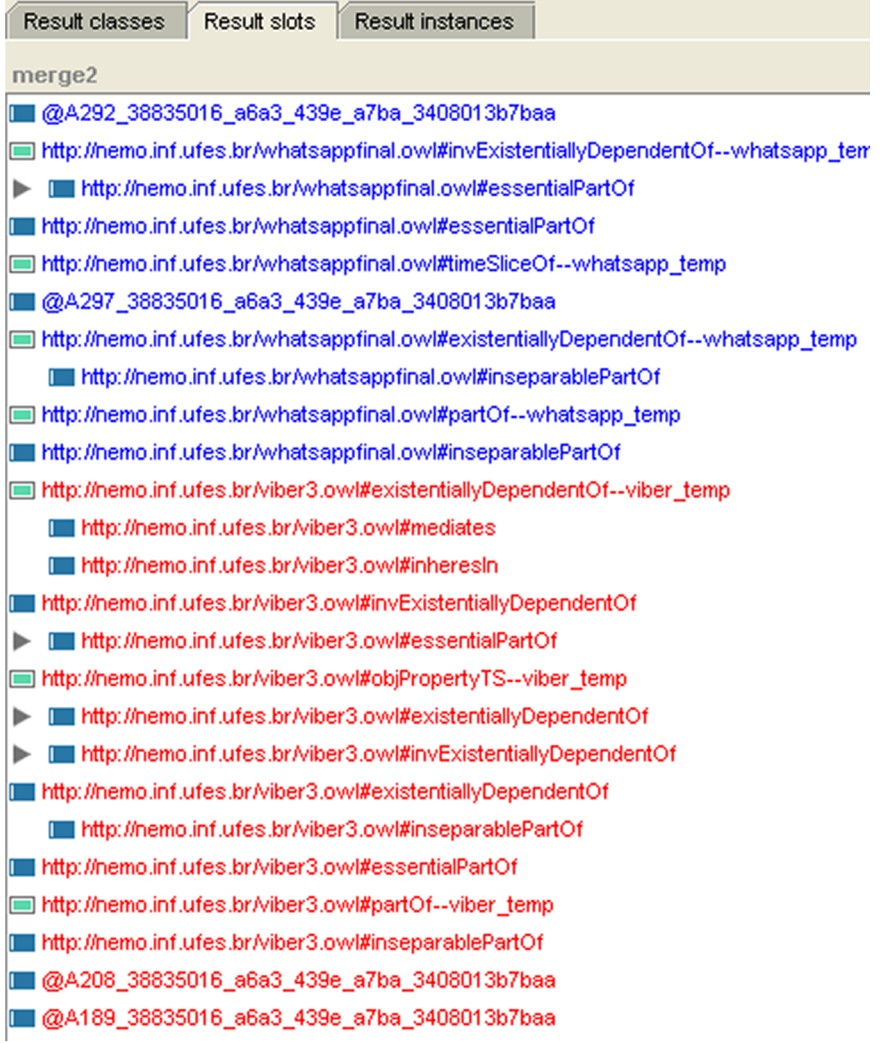

**Figure 5 The result slots of the ontology after integration in Protégé.**

is proposed in this article for the evaluation of the proposed method. The case study is downloaded from the APKMirror. The "Avast Android Mobile Security" secures the devices against phishing attacks from emails, phone calls, infected websites, or SMS messages. Also, it has many other features as Antivirus Engine, App Lock, Call Blocker, Anti-Theft, Photo Vault, virtual private network, and Power Save. The reason for choosing "the Avast Android Mobile Security" application as a case study is that it has the maximum number of the detected anti-patterns using the proposed method. Using the reverse methodology, we generated the UML class diagram model of the Java classes in Modelio. The model includes the classes, subclasses, class attributes, operations, and the associations between them (Fig. 6).

After generating the UML class diagram of the application in Modelio, we detected 229 repeated anti-patterns in the "Avast Android Mobile Security." The anti-patterns are shown in Fig. 7. The number and the location of the anti-patterns were determined.

**Table 1 The description of the mobile apps under analysis.**

| Mobile application name | Size (MB) | Downloads | Description of use |
|---|---|---|---|
| Test DPC 4.0.5 | 3.14 | 1,076,791 | Libraries and demo |
| Avast 6.5.3 Security | 20.71 | 1,364 | Antivirus engine and mobile security |
| Free-Calls-Messages | 31.59 | 1,537 | Communication |
| Beautiful Gallery 2.3 | 11.31 | 497 | Photography |
| Play Store 9.3.4 | 14.17 | 6,950 | Google Play Store |
| Wall Paper 1.2.166 | 2.29 | 9,730 | Personalization |
| Oasis-Feng/Island 2.5 | 2.34 | 822 | Privacy protection and parallel running |
| Netflix-5-4-0-Build | 18.81 | 22,043 | Entertainment |
| Remainder 1.4.02 | 9.36 | 3,612 | Remainder |
| Sound-Picker 8.0.0 | 3.9 | 2,142 | Samsung sound picker |
| Air-Command 2.5.15 | 0.82 | 1,747 | Air command |
| Lifesum-Healthy-Lifestyle | 31.4 | 3,594 | Diet plan, food diary, macro calculator, calorie counter, and healthy recipes |
| Background-Defocus 2.2.9 | 3.45 | 2,960 | Photography |
| Gasbuddy-Find-Cheap-Gas | 29.64 | 334 | Travel and local |
| Soundcloud-Music-Audio.03.03 | 33.2 | 2,066 | Music and audio |
| Network-Monitor-Mini 1.0.197 | 2.88 | 307 | Monitor the upload and download speed per second |
| Casper Android 1.5.6.6 | 18.77 | 383,765 | Messaging app snapchat |
| Line 8.4.0 | 70.25 | 260 | Communication |
| Diagnosises | 6.96 | 36 | Medical |
| Viber 7.7.0.21 | 38.4 | 1,628 | Communication |
| WhatsApp 2.17.235 | 35.81 | 28,978 | Communication |
| Firefox 56.0 | 40.62 | 20,423 | Communication |
| Blue- Email and Calendar 1.9.3.21 | 43.2.4 | 203 | Productivity |
| Google Camera 5.1.011.17 | 36.48 | 211,822 | Photography |
| YouTube 13.07 | 24.13 | 23,667 | Video players |
| True Caller 8.84.12 | 23.09 | 609 | Communication |
| Samsung Gallery 5.4.01 | 17.61 | 10,712 | Photography |
| Twitter 7.48.0 | 35.82 | 694 | News and magazines |
| Chrome Browser 66.0.3359 | 41.51 | 29,129 | Communication |

There were 10 detected anti-patterns (without repeat): "NameSpaces have the same name," "NameSpace is Leaf and is derived," "NameSpace is Leaf and is abstract," "Generalization between two incompatible elements," "A public association between two Classifiers one of them is public and the other is privet," "Classifier has several operations with the same signature," "Classifier has attributes with the same name," "The status of an Attribute is abstract and class," "A destructor has two parameters," and finally "MultiplicityMin must be inferior to MultiplicityMax." Figure 8 shows a sample of them.

To convert the UML model to XML format, we converted it into an enterprise architecture file then converted it to an OLED file. In the "Avast Android Mobile Security" OLED file, we validated the model for detecting the anti-patterns. The detected

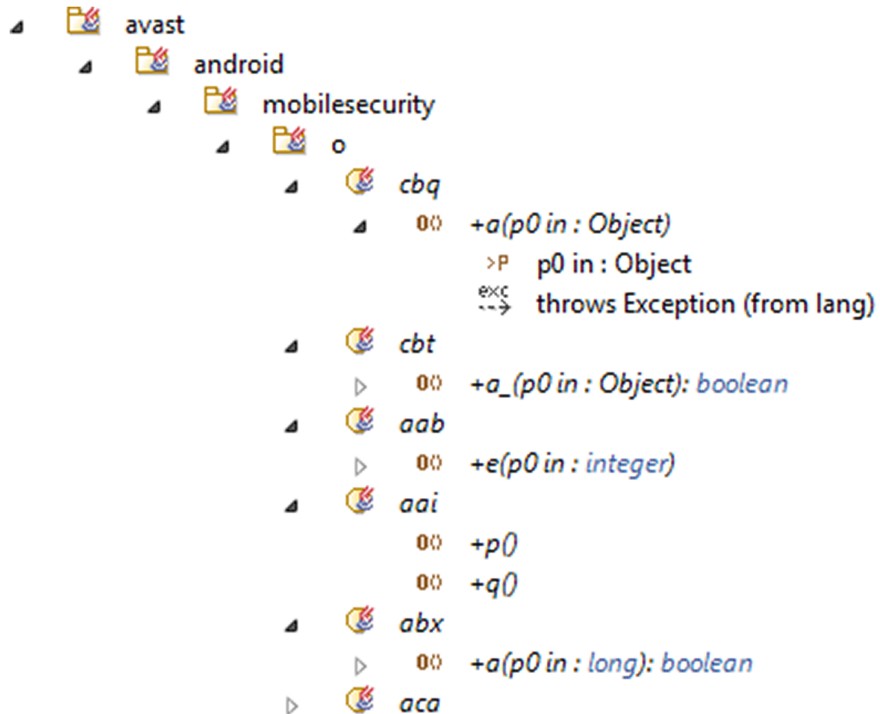

**Figure 6 The generated UML class diagram of the case study.**

| | | |
|---|---|---|
| a | R1980 | The Classifier 'a' has at least two Attributes or two AssociationEnds with the |
| ViewDecorator_ | R2260 | The Classifier 'ViewDecorator_Factory' has several operations with the same |
| a | R2260 | The Classifier 'a' has several operations with the same signature. |
| bxo | R2060 | There are several namespaces with the same name in the namespace 'bxo'. |
| ra | R2260 | The Classifier 'ra' has several operations with the same signature. |
| a | R2260 | The Classifier 'a' has several operations with the same signature. |
| ProviderOfLazy | R2260 | The Classifier 'ProviderOfLazy' has several operations with the same signatu |
| a | R2260 | The Classifier 'a' has several operations with the same signature. |
| b | R2260 | The Classifier 'b' has several operations with the same signature. |
| a | R2260 | The Classifier 'a' has several operations with the same signature. |
| Buffer | R2260 | The Classifier 'Buffer' has several operations with the same signature. |
| il | R2260 | The Classifier 'il' has several operations with the same signature. |
| o | R2260 | The Classifier 'o' has several operations with the same signature. |
| StatementExec | R2260 | The Classifier 'StatementExecutor' has several operations with the same sig |
| f | R2260 | The Classifier 'f' has several operations with the same signature. |
| c | R2260 | The Classifier 'c' has several operations with the same signature |

**Figure 7 Modelio anti-patterns.**

anti-patterns in the different apps were: association cycle anti-patterns, Binary relations with overlapping ends anti-patterns, imprecise abstraction anti-patterns, and relation composition anti-patterns.

After anti-patterns detection using OntoUML editor, OLED supports the transformation of OLED file to the OWL ontology. We checked the inconsistency anti-patterns using the reasoner of the ontology editor (Protégé). The reasoner detected the anti-patterns

**Figure 8 The anti-pattern "Classifier has several operations with the same signature."**

related to inconsistency as (similar name, multiplicity constraints, and cyclic inheritance). Using the reasoner of ontology over the case study, we detected the anti-patterns in the classes that have the anti-patterns NameSpaces have the same name, classifier has several operations with the same signature, classifier has attributes with the same name, and MultiplicityMin must be inferior to MultiplicityMax, which we detected after generating the class diagram in Modelio, and detected the anti-pattern (association cyclic) which was detected via OLED.

The treatment or correction of the detected anti-patterns is classified into the following:

- Modelio presents the solution as a list of recommendation which developer can do it manually. In this case study, Table 2 presents the anti-patterns and the method of correction.
- OLED presents automatic solutions to correct the anti-patterns which we list in Table 3.
- Reasoner in Protégé presents all inconsistency anti-patterns where as Reasoner gives just the location of the inconsistent classes as in Fig. 9.

## RESULTS AND DISCUSSION

We applied our proposed method on a sample of 29 Android applications, which we downloaded from the APK Mirror. The results present the detected anti-patterns in the 29 mobile applications and the relation between the different types of anti-patterns. The proposed method detected 15 anti-patterns. The total number of anti-patterns that appeared in the 29 applications was 1,262 anti-patterns. We classified the anti-patterns according to their existence in the UML class diagram components. The occurrences of the anti-patterns are given in Table 4. Every group has the anti-patterns that were detected in

**Table 2  Ten Modelio anti-patterns and their correction way.**

| The anti-pattern | The method of correction |
|---|---|
| NameSpaces have the same name | Change the name of the conflicting *NameSpaces* |
| NameSpace is Leaf and is derived | Make the *NameSpace* non-final |
| NameSpace is Leaf and is abstract | Make the NameSpace non-final |
| Generalization between two incompatible elements | Change the source or the target in order to link two compatible elements |
| A public association between two Classifiers one of them is public and the other has different visibility | Change the visibility of the target class to public |
| Classifier has several operations with the same signature | Rename one of the *Operations* or change their parameters |
| Classifier has attributes with the same name | Rename the *Classifiers Attributes* |
| MultiplicityMin must be inferior to MultiplicityMax | Change the value of the minimum multiplicity to be less than the maximum multiplicity |
| The status of an Attribute is abstract and class at the same time | Set only one of the statuses to true |
| A destructor has parameters | Remove these parameters or remove the destructor stereotype from the method |

**Table 3  OntoUML anti-patterns and the correction way.**

| The anti-pattern | The method of correction |
|---|---|
| Association cycle | Chang the cycle to be closed or open cycle |
| Binary relation with overlapping ends | Declare the relation as anti-reflexive, asymmetric, and anti-transitive |
| Imprecise abstraction | Add domain-specific constraints to refer to which subtypes of the association end to be an instance of the other end may be related |
| Relation composition | Add OCL constraints which guarantee that if there is a relation between two types and one of them has subtypes, there must be constraints says that the subtypes are also in a relation with the other type |
| Relation specialization | Add constraints on the relation between the type and the super-type, declaring that the type is to be either a specialization, a subset, a redefinition or disjoint with relation SR |

it. For example, the group "Anti-patterns in Operations" presents all anti-patterns that were detected in the operations using the three tools.

Table 5 shows the detected anti-patterns in each application using the proposed method and the total number of anti-patterns in the 29 mobile applications.

We found that the "anti-patterns in the class" group is the most commonly detected anti-pattern in Android applications. The "anti-patterns in operation" is the least commonly appeared anti-pattern (Fig. 10).

We measured the relations between anti-patterns groups using correlation coefficient. Correlation coefficient is a statistical measure of the degree to which changing the value of one variable predict changing to the value of the other. A positive correlation indicates that the extent to which those variables increase or decrease in parallel. While a negative correlation indicates the extent to which one variable increases as the other

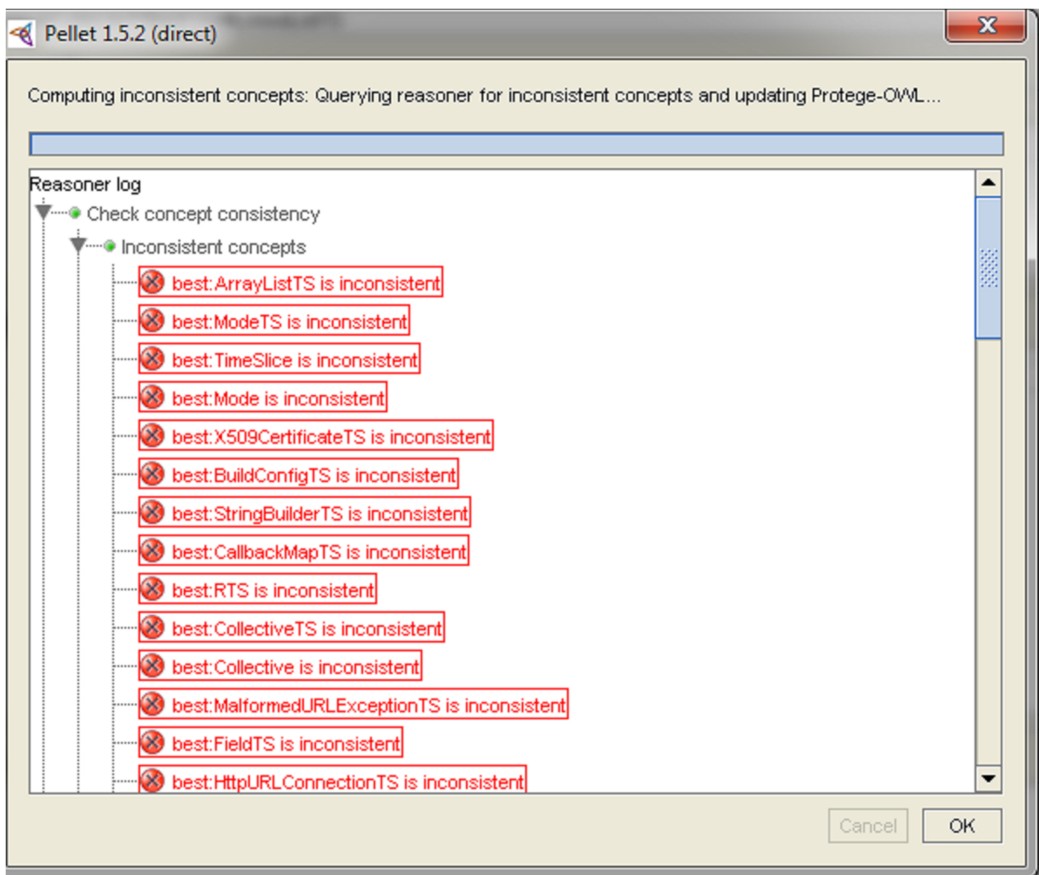

**Figure 9 The inconsistent classes using reasoner detection.**

**Table 4 Occurrences of the anti-patterns in the mobile apps.**

| The group | Percentage of occurrences across models | Total # of occurrences |
| --- | --- | --- |
| Anti-patterns in attributes | 0.713% | 9 |
| Anti-patterns in namespaces | 7.210% | 91 |
| Anti-patterns in operations | 0.396% | 5 |
| Anti-patterns in associations | 43.898% | 554 |
| Anti-patterns in the class | 47.78% | 603 |
| Total | | 1,262 |

decreases. Table 6 presents the correlations between anti-patterns groups. The tool can detect certain group, it also can detect in parallel the other as attributes anti-patterns with operations anti-patterns. Also, appearance of attributes anti-patterns in certain applications indicates the appearance of operations anti-patterns strongly. Then the correlation between the five groups of anti-patterns is used to know if the existence of any type of them implies the existence of other type. There was a strong negative correlation (−0.1) between namespaces anti-patterns and association anti-patterns. Also, a strong positive correlation (0.8) between attributes anti-patterns and operations anti-patterns.

**Table 5 The anti-patterns in each app.**

| | Mobile app | CHSO | NHSN | NLAD | NLAA | GBUE | CHSA | MMITMM | PACPP | SAAC | TDHPS | BinOver | AC | RS | RelComp | ImpAbs | Total |
|---|---|---|---|---|---|---|---|---|---|---|---|---|---|---|---|---|---|
| 1 | Test DPC 4.0.5 | 7 | 2 | 1 | – | – | 2 | – | 1 | – | – | 10 | 6 | 1 | – | – | 30 |
| 2 | Avast Android Mobile Security | 149 | 15 | 2 | 2 | – | 58 | 3 | – | 2 | – | 6 | – | – | 2 | 3 | 240 |
| 3 | Free-Calls-Messages | 4 | 1 | 2 | – | 2 | 1 | 1 | – | – | – | 3 | 2 | – | – | – | 16 |
| 4 | Beautiful Gallery 2.1 | 5 | 2 | – | – | – | – | – | 1 | – | – | – | – | – | – | – | 8 |
| 5 | Play Store | 8 | 1 | 1 | – | 1 | 1 | 2 | 3 | 1 | – | 6 | 16 | – | 41 | 2 | 82 |
| 6 | Wall Paper | 1 | 1 | – | – | – | – | 1 | – | – | – | – | – | – | 4 | – | 7 |
| 7 | Oasis-Feng/Island | 17 | 2 | – | – | – | 4 | – | – | – | – | – | – | – | – | 3 | 26 |
| 8 | Netflix-5-4-0-Build | 60 | 7 | 2 | 2 | – | – | – | – | – | 5 | 5 | – | – | – | – | 79 |
| 9 | Remainder | 11 | 4 | – | – | – | 1 | 2 | – | – | – | 4 | 7 | 5 | – | – | 34 |
| 10 | Sound-picker | 9 | 1 | – | – | – | – | – | – | – | – | – | – | – | – | 2 | 12 |
| 11 | Air-Command | 8 | 1 | – | – | 1 | – | – | – | – | – | – | – | – | – | – | 10 |
| 12 | Lifesum-Healthy-Lifestyle | 5 | 1 | – | – | – | – | 1 | – | 4 | – | – | 5 | 1 | 2 | 2 | 21 |
| 13 | Background-Defocus | 10 | 4 | – | – | – | 4 | 1 | – | – | – | 10 | – | – | 6 | – | 35 |
| 14 | Gasbuddy-Find-Cheap-Gas | 11 | 4 | – | 1 | – | 2 | – | – | 1 | – | – | 7 | 2 | – | 3 | 31 |
| 15 | Soundcloud-Music-Audio | 6 | 4 | – | – | – | – | – | 2 | – | – | – | – | 8 | 1 | 2 | 23 |
| 16 | Network-Monitor-Mini | 7 | 2 | – | – | – | 1 | 2 | – | – | – | – | 3 | – | – | – | 15 |
| 17 | Casper Android | 6 | 4 | – | – | – | – | – | 3 | – | – | 20 | – | 6 | – | – | 39 |
| 18 | Line | 15 | 1 | – | – | – | 1 | 1 | – | – | – | – | 6 | – | 2 | 1 | 27 |
| 19 | Diagnoses | 1 | – | – | – | – | – | – | – | – | – | – | 2 | 1 | – | – | 4 |
| 20 | Viber | 42 | 4 | – | – | 1 | 1 | – | – | 1 | – | 9 | – | 7 | 5 | – | 69 |
| 21 | WhatsApp | 5 | 1 | – | – | – | – | 2 | – | – | – | 30 | – | 2 | – | 2 | 42 |
| 22 | Firefox | 40 | 4 | – | – | – | 1 | 1 | 4 | – | – | – | – | 8 | – | 1 | 59 |
| 23 | Email and Calendar | 15 | 2 | – | – | – | 1 | – | – | – | – | 108 | 2 | – | – | – | 128 |
| 24 | Google Camera | 9 | 1 | – | – | – | – | – | 1 | – | – | 15 | 8 | – | 1 | 1 | 36 |
| 25 | YouTube | 21 | 4 | – | – | – | 3 | – | – | – | – | 3 | 3 | – | 3 | 2 | 39 |
| 26 | True Caller | 31 | 2 | – | – | – | – | 2 | – | – | – | 17 | 5 | – | 1 | – | 58 |
| 27 | Samsung Gallery | 12 | – | – | – | – | – | – | 1 | – | – | – | 9 | 3 | – | 1 | 26 |
| 28 | Twitter | 6 | 2 | – | – | – | – | 1 | – | – | – | 15 | 6 | 1 | 1 | – | 32 |
| 29 | Chrome browser | 1 | 4 | 1 | – | – | – | – | – | – | – | 9 | 5 | – | 12 | – | 32 |
| | # of appearance | 522 | 81 | 5 | 5 | 5 | 81 | 20 | 16 | 9 | 5 | 270 | 92 | 45 | 81 | 25 | 1,262 |

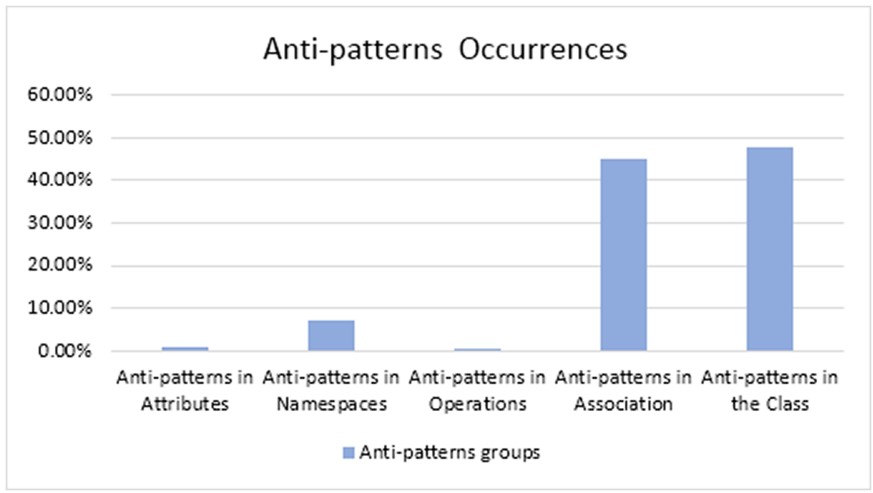

**Figure 10 The occurrences of the detected anti-patterns' groups.**

**Table 6 The correlation among anti-patterns groups.**

| Anti-patterns | Correlation coefficient ($r$) |
| --- | --- |
| Attributes and namespaces | −0.049 |
| Attributes and operations | 0.884 |
| Attributes and associations | 0.196 |
| Attributes and classes | 0.342 |
| Namespaces and operations | −0.060 |
| Namespaces and associations | −0.121 |
| Namespaces and classes | 0.010 |
| Operations and associations | 0.345 |
| Operations and classes | 0.267 |
| Associations and classes | 0.070 |

Also, we analyzed the correlation between the detection tools of the proposed method (Table 7). The greatest correlations were between Modelio and Protégé. For assessing the direct relation between Protégé and Modelio, we calculated the statistical means of anti-patterns which were detected by each tool (Modelio, Protégé, and OLED) on 29 mobile applications as in Fig. 11. Figure 11 shows the similarity between both the means of Protégé and Modelio as the result of the correlation. Now, we want to statistically answer the question "Do we need to use the three tools" and "is there a relation between them?"

In order for statistical analysis to explain the relation among the three tools and the anti-patterns' groups, we used the analysis of variance ANOVA test. This is to determine whether there are any statistically significant differences between the means of anti-patterns detection by each one of the tools, and also to determine if there is any relation between anti-patterns groups and the features of mobile applications.

**Table 7 The correlation among the three tools.**

| Systems | Correlation coefficient ($r$) | Specification |
|---|---|---|
| Modelio and OntoUml | −0.032 | There is a reverse correlation between Modelio and OntoUml |
| Modelio and Protégé | 0.966 | There is a direct correlation between Modelio and Protégé |
| Protégé and OntoUML | −0.060 | There is a reverse correlation between Protégé and OntoUml editor |

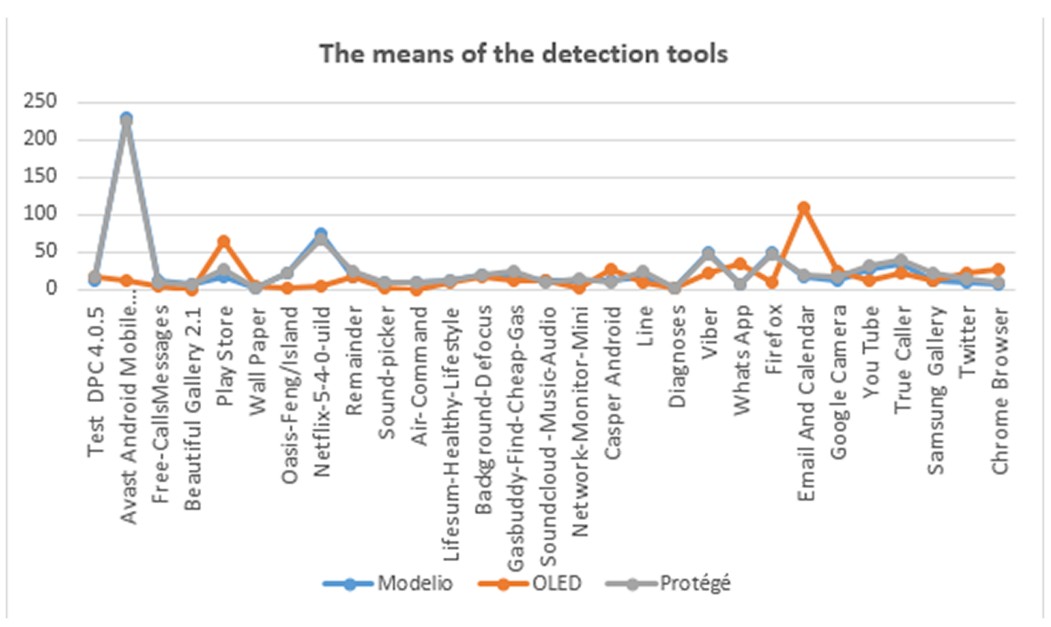

**Figure 11 The means of the detected tools.**

We use ANOVA to calculate a test ($F$-ratio) with which we can obtain the probability $P$-value (usually taken as $P < 0.05$) suggests that at least one group mean is significantly different from the others. The null hypothesis is (all population means are equal). The alternative hypothesis is (at least one population mean is different from the rest). Where the degree of freedom (d$f$) between groups is 28 and d$f$ within the group is 116. We found that the significant differences are 0.578, 0.464, and 0.926 for Protégé, Modelio, and OLED, respectively. This implies that the null hypothesis is false, i.e., all the detection tools are necessary and required for the detection of the anti-patterns.

The ANOVA statistically proved that there was no concern for the features or the specifications of the applications; that is, the low $F$-value meant that the groups are close together relative to the variability within each group.

We separated the result of integration phase because it is an optional phase. In the case of homogeneous applications, we found that the number of the detected anti-patterns in the output application was not the same. The detected anti-patterns using the ontology integration tool Prompt was less than the number of anti-patterns detected by using the

**Table 8 Anti-patterns number before and after merging.**

| Mobile apps | Viber | WhatsApp | The integrated app | Total |
|---|---|---|---|---|
| (Merging UML designs) # of detected anti-patterns in first method using Modelio | 49 | 8 | 58 | 115 |
| (Merging ontologies) # of detected anti-patterns in second method using Protégé | 51 | 64 | 13 | 128 |

Modelio tool. This indicates that semantic integration decreases the increases the accuracy of detecting anti-patterns in mobile applications. Table 8 shows the number of anti-patterns in each application in the integration case study (Viber and WhatsApp) and the number of them in the mobile application pattern after merging. The enhancement using ontology is approximately by 11.3% in addition to a consistency check.

Where the formula to calculate the increasing percent between two values is

$$\text{Percent increase} = \left[ \frac{(\text{Second value} - \text{First value})}{(\text{First value})} \right] \times 100 \qquad (1)$$

Substitute in Eq. (1) by

The first value is the total number of anti-patterns according to using Modelio = 115.

The second value is the total number of anti-patterns according to using Prompt = 128.

Then the percent is increased by $\cong$ 11.3% which implies that using ontology integration by Prompt (Protégé plugin) instead of using UML integration by Modelio increases the percent of detection.

Additionally, using ontology to separately refine Viber or WhatsApp as a pattern enhanced them approximately 4.04% and 89%, respectively, in addition to a consistency check by "Reasoner."

## CONCLUSIONS

In this paper, we focused on improving the quality of mobile applications. We introduced a general method to automatically detect anti-patterns not by using specific queries, but by using Modelio, OLED, and Protégé in a specific order to get positive results. Also, concerning the related work section, our proposed method is more general than other methods as the proposed method supports semantic and structural anti-pattern detection at the design level.

For evaluation of the proposed method, we applied it on a sample of 29 mobile applications. It detected 15 semantic and structural design anti-patterns. According to the proposed classification of anti-patterns, "the anti-patterns in the class group" was the most frequent anti-pattern, and "the anti-patterns in the attribute group" was the least frequent. From the perspective of anti-patterns detection, the analysis of results also showed that there is a correlation between the Modelio and Protégé platforms. Also, there is no correlation between OLED and Protégé and no correlation between Modelio and OLED.

We found that using ontology in the integration phase increases the detection percentage approximately by 11.3% and guarantees consistency which is assessed by the reasoner of the ontology. Accordingly, semantic ontology integration has a positive effect on the quality of the new application. This helped with developing a correct, consistent, and coherent integrated pattern that has few anti-patterns.

Finally, we recommend that the developer, before using any mobile application as a pattern, should check the design of the selected application against the anti-pattern.

When a developer concerned with avoiding certain anti-patterns type, the correlations between anti-patterns groups, and between tools will help him. Also, the proposed method considered the issues and problems of developers who are revising Android applications and integrating new packages of code skill sets. A code review such as the methodology proposed could be very valuable in terms of not carrying forward existing anti-patterns and not incorporating new code flawed with poor design. The reverse deeply in OWL ontology of a mobile application very useful.

In the future, we are going to solve the problem of big ontologies which cannot be opened in ontology editors as Protégé to complete the detection process. Although, detection of anti-patterns at the design level is very useful and reduces some anti-patterns in the code level, we will refine the metric method for detecting code level anti-patterns on big ontology. Also, we will create a semantic web application for anti-patterns to collect all detection tools of the two levels and anti-patterns catalog. Finally, the correction phases in Modelio and Reasoner are still open issues.

### Funding
The authors received no funding for this work.

### Competing Interests
The authors declare that they have no competing interests.

### Author Contributions
- Eman K. Elsayed conceived and designed the experiments, performed the experiments, contributed reagents/materials/analysis tools, analyzed the data, prepared figures and/or tables, performed the computation work, authored or reviewed drafts of the paper, approved the final draft
- Kamal A. ElDahshan authored or reviewed drafts of the paper, approved the final draft.
- Enas E. El-Sharawy conceived and designed the experiments, performed the experiments.
- Naglaa E. Ghannam conceived and designed the experiments, performed the experiments, contributed reagents/materials/analysis tools, analyzed the data, prepared figures and/or tables, performed the computation work, authored or reviewed drafts of the paper.
## Data Availability

The features of the downloaded mobile apps and the detected anti-patterns are available as a Supplemental File. The file shows the relation between the detected anti-patterns and the detection tools and the relation between anti-patterns groups and the detection tools.

## Supplemental Information

Supplemental information for this article can be found online at http://dx.doi.org/10.7717/peerj-cs.212#supplemental-information.

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
