# Peer review of "Reverse engineering approach for improving the quality of mobile applications"

_PeerJ Computer Science, doi:10.7717/peerj-cs.212_

## Round 0.1 · original submission · Minor Revisions

The paper requires some major revisions. I think that the reviewers suggestions should be very useful to improve the paper quality.

Reviewer 1 ·

Basic reporting

1) There are some typos and some sentences that are not clear as highlighted in red in the document
2) Some sentences need references and some context is not relevant as mentioned in the attached document
3) The paper lacks organization and structure and the figures and tables are not well formatted and need more description
4) not clear
5) results are not clear

Experimental design

1) The paper includes case studies about Android Applications and how to improve the anti-design patterns in them so it may not be within the scope of the journal

2) Not clear

3) Not clear

4) Yes

Validity of the findings

1) Novelty is limited and not highlighted properly

2) Somewhat

3) Not clear

4) __

Annotated reviews are not available for download in order to protect the identity of reviewers who chose to remain anonymous.

·

Basic reporting

Abstract Editorial Comments

Methods:
1) Include the decision criteria used to select the 29 apps used in testing and source of downloads. Include what version of Android was used. What was the criteria? Why did you select these apps? The list of apps tested includes two browsers (Firefox and Chrome) which have significantly different characteristics than some of the other apps.
2) While the method of reverse engineering and analyzing the Android apps does appear to be novel and potentially provide for improvement in app design and app revisions, it does not necessarily mean that it will “lead the way for perfect long-time apps and ensure that the applications are purely valid.” Rather, the authors might want to suggest that this methodology provides developers with a way to help ensure that anti-patterns in existing apps are not repeated in-app revisions and may improve both operational characteristics and user experience.
3) The authors should state that they present and test a new method of generating OWL Ontology of mobile apps and analyze the relationships among object-oriented anti-patterns and offer methods to resolve the anti-patterns by detecting and treating 15 different design’s semantic and structure anti-patterns that occurred in analyzing the 29 apps.

Results: The authors may want to combine the first two sentences into something like this. “The proposed method is a general detection method to identify and help resolve semantic and structural design anti-patterns in Android mobile apps. The analyses of the various types and frequency of anti-patterns and the potential uses of the demonstrated methodology in integrating the mobile applications are shown in order to validate the approach. “

General Editorial Comments:
1) At first, the paper appears to be well structured with the sections on Abstract, Related Work, Ontology and Software Engineering, Method (not proposed if it was actually accomplished).
2) However, what is unclear is why the authors chose to include the “Integration of Mobile Apps.” Most app integration is for the inclusion of new skill sets for the app such as IOT or monitoring applications or potentially voice-activation integration into an existing app. Seldom would we anticipate integrating 2 separate apps? While the diagrams in Figures 2 and 3 are interesting, this entire section seems out of place and fails to establish the logic what apps are being combined and why? Is this the only way to test and validate the methodology to identify and resolve anti-patterns?
3) The Implementation section is well written and the descriptions appear clear for the re-engineering of an app and the detection of the anti-patterns in the UML class diagram, with the eventual editing of the OWL ontology of a single app using the reasoner ontology the Protégé editor. Again, this begs the question of why the discussion of the integration of apps (Line 161). Unless, the authors are really addressing current needs, not of integrating two different apps, but rather the integration of new modules of skill sets into the apps? (Note: Further reading in Empirical Validations shows that the authors did integrate “Viber” and “What’sApp” as a case study. But no rationale is provided for why this integration is needed to demonstrate the methodology for identifying and resolving anti-patterns.

Experimental design

4) The Empirical Validation Section begins well with the statement that the authors propose to “reporting on the results…for the detection of 15 anti-patterns on the 29 popular Android apps downloaded from APK Mirror.”
5) The Case Study is a very confusing section. The authors first state, “ To explain the proposed method we presented a snapshot in a different case study, “ A vast Android Mobile Security.” This case study is not explained nor cited. Where did it come from? Why is it being used now?
6) Actually, with some explanation of what it is the authors are examining (which is entirely unclear) this section has some very useful information for developers as it shows the number and locations and types of anti-patterns that were found in whatever mysterious app source the authors are exploring. If this is a common pattern in Android app design development, then the identification of the resulting anti-patterns is very helpful. The problem is the reader has no idea what app is being analyzed. This is a major flaw in the article.

Integration of Mobile Applications
1) Once again, the reader is mystified by what exactly is being explained and why the authors chose to integrate Viber and WhatsApp. The paper would likely be much stronger without this section or this analysis unless the authors can provide a rationale for the integration and provide an analysis that tells the reader more about the potential use of the proposed methodology.

Discussion of review of Tables and Charts
The authors should review each of the tables for alignment and correctness of the information and identification of data. For example, in Table 3, the heading might read “The Method of Correction.” Check for placement of % marks, as they should always follow the numbers rather than precede. In Table 4 the alignment of the second column is confusing. Also, in Table 4 what is “The Group”? Is it the 29 Apps Under Review?

Validity of the findings

RESULTS and DISCUSSION
The results and analyses of the 29 apps and the resulting 29 anti-patterns and their classifications are the key results of this section (Please see editorial notes on Tables and Figures). There is a typo in line 256 (it should read 29, not 39). There should be additional discussion regarding any thoughts about the strength in numbers of the various anti-patterns shown in Table 9. Why, for example might we anticipate that “anti-patterns in class” at ~50% is the most commonly detected? But “anti-patterns in association” is nearly as common at ~45%. The other 3 anti-patterns are detected at much lower levels. What can developers and team architects learn from this analysis?
The correlation discussion should include both negative and positive correlations ( remember a correlation of -1) is a strong correlation and should discuss any correlation over +/- .3 . In Figure 9 we learn that “Attributes in Class” and “Attributes in Operations,” the areas with the greatest correlation also occur less than ~1% of the time. So, what is this telling us? How is this helpful?
The discussion of the ANOVA test among the three tools used sequentially in the proposed methodology and the detected anti-patterns is interesting. What is not clear from the discussion is that the analyses were proposed to be done as a comparative, rather it appeared that the methodology being proposed was a step-by-step method.

Comparing the Mobile App Integration Methods
Here again, the logic of the paper fails and confusion sets in. What 2 methods in 6.3? Is this part of another paper? When did the single methodology proposed become 2 methods being compared-- Ontology Integration and UML? What is the virtue or benefit of integrating Viber and WhatsApp? What does it mean when the authors state that “semantic integration decreases the accuracy of the anti-patterns in apps??” It appears that using ontology separately does increase the ability to identify and potentially improve the coding and structure of the apps? But how does integration help? This is very confusing

Discussion
There was no discussion section. Perhaps it would benefit the authors and potential future readers to identify the key findings and support them as building upon prior research and science. How do these results support prior work? How are they different?
What strengths and limitations exist in this work (after the paper is cleaned up…) Were there any unexpected outcomes or learning?

Conclusions
It is suggested that the authors closely consider the concerns and suggestions of this reviewer and others to re-construct this article. The methodology proposed is indeed interesting and the potential for using ontology and the existing tools in the order proposed may indeed be helpful for app developers, especially those who are in a revision process. The authors should briefly restate their hypotheses and expectations. (This suggests a close review of the list from lines 65-77). The benefits and discussion of the 2 app integration should be closely considered for value. What would perhaps be more beneficial would be a comparison of code between an existing app and any package extensions being considered, for example for IOT or voice control integration. As a stand-alone analysis it is sufficient to discuss the 29 apps and the presence and types of anti-patterns found. However, this requires a discussion and rationale for the selection of the apps—more than just that they are “popular.”
Conclusions should also synthesize what was really learned in the analysis and how can this help developers build and revise apps in a cleaner fashion by removing and correcting anti-patterns. What are any supporting research results? What are next steps going forward to further support or analyze this methodology.



Discussion of review of Tables and Charts
The authors should review each of the tables for alignment and correctness of the information and identification of data. For example, in Table 3, the heading might read “The Method of Correction.” Check for placement of % marks, as they should always follow the numbers rather than precede. In Table 4 the alignment of the second column is confusing. Also , in Table 4 what is “The Group”? Is it the 29 Apps Under Review?

Additional comments

There is significant potential in this proposed methodology and the research demonstrates the value of the step-by-step method and the use of ontology in identifying and correcting anti-patterns.
However, the article as written is confusing and needs significant corrections and re-thinking of the overall structure. Especially confusing is the integration of the 2 apps Viber and WhatsApp. Why? Rather the authors should consider the issues and problems of developers who are revising Android apps and integrating new packages of code skill sets. A code review such as the methodology proposed could be very valuable in terms of not carrying forward existing anti-patterns and also not incorporating new code flawed with poor design.

Reviewer 3 ·

Basic reporting

It would be useful to have improved consistency in Tables 2-6. E.g. Tables 2 and 3 name specific anti-patterns; whereas Tables 4 and 6 list groups of anti-patterns (without explicitly linking specific items to these groups); and Table 5 uses abbreviations.

Also it is difficult to follow Table 5 - if allowed by the journal style guide it might be easier to read with vertical column lines, or dashes ("-") rather than blanks for empty cells.

Figure 10 would be better presented as a Table. It was not clear what is being shown in Figures 7 and 8.

The language is generally clear, but could be improved throughout. Some specific cases where improvements are needed: e.g. in the abstract, "mind-bloggling programming frameworks" could be written more professionally. "... leading the way for perfect long-term apps and ensuring that these applications are purely valid" - it is perhaps too much to say that this research will lead to "perfect apps".

Line 273 refers to "Section 6.3" - which is perhaps from a different version of the manuscript?

Line 276: "semantic integration decreases the accuracy of anti-patterns in apps" - do the authors mean "decreases the occurrence of anti-patterns"? (anti-patterns do not have accuracy).

Experimental design

Line 199 "We selected some popular apps" - was there a particular selection process? Why were these apps chosen?

Line 219 "we introduce... the correction method" - please clarify whether these are standard correction methods, or new methods that the authors are introducing in this manuscript?

Lines 243-247 - I was not clear on the rationale for integrating/combining two separate apps (e.g. WhatsApp and Viber) - perhaps the authors could clarify this?

Line 260-262 - the authors say they perform correlations, but it is not clear what is being correlated. In each case, is it the total number of occurrences? The rationale for these correlations was also not clear.

Validity of the findings

Lines 263-264 - "the greatest correlations were between attributes and operations groups" - however these had the smallest numbers - this should be acknowledged.

Lines 266-270 - the rationale for the ANOVA is not clearly stated - but I also don't understand the interpretation of these results. If there is no significant difference associated with any specific tool (p values are >> 0.05), does that mean we can use any *one* tool, rather than having to use them all?

Lines 278-280 - it is not clear how the % values are obtained. Also, these refer to "consistency checks" which are not described elsewhere.

Line 294 "there is a correlation between OLED and Protege" - Table 7 shows an r value of -0.060 which is negligible. Inferring a correlation from this result is unfounded.

Line 297: "guarantees consistency" - it is not clear from these results what is meant by consistency, and how this can be guaranteed.

Additional comments

As a general comment, I think this manuscript would benefit from a discussion on the relationship between the reverse engineering process and the anti-pattern detection process. Is there any consideration of:
i) does the reverse engineering process (in general, or for these specific apps) have risks associated with the quality of source code obtained?
ii) do your findings tell us anything specifically about this set of apps?
iii) would the process work as well if source code was obtained directly (e.g. from GitHub) rather than the first reverse engineering step?

---

## Round 0.2 · accepted · Accept

The paper can be published

·

Basic reporting

The authors have sufficiently addressed my questions and concerns in their rebuttal and revision of the article.

Experimental design

no comment

Validity of the findings

No comment